# Implementation of KEIGAAF in Primary Schools: A Mutual Adaptation Physical Activity and Nutrition Intervention

**DOI:** 10.3390/ijerph17030751

**Published:** 2020-01-24

**Authors:** Sacha R.B. Verjans-Janssen, Sanne M.P.L. Gerards, Anke H. Verhees, Stef P.J. Kremers, Steven B. Vos, Maria W.J. Jansen, Dave H.H. Van Kann

**Affiliations:** 1Department of Health Promotion, NUTRIM School of Nutrition and Translational Research in Metabolism, Maastricht University, 6229 HA Maastricht, The Netherlands; 2Department of Industrial Design, Eindhoven University of Technology, 5612 AZ Eindhoven, The Netherlands; 3School of Sport Studies, Fontys University of Applied Sciences, 5644 HZ Eindhoven, The Netherlands; 4Academic Collaborative Center for Public Health, Public Health Service South-Limburg, 6400 AA Heerlen, The Netherlands; 5Department of Health Services Research, Maastricht University, CAPHRI Care and Public Health Research Institute, 6229 GT Maastricht, The Netherlands

**Keywords:** school health promotion, implementation, context, adaptation, qualitative

## Abstract

School health promotion is advocated. Implementation studies on school health promotion are less often conducted as effectiveness studies and are mainly conducted conventionally by assessing fidelity of “one size fits all” interventions. However, interventions that allow for local adaptation are more appropriate and require a different evaluation approach. We evaluated a mutual adaptation physical activity and nutrition intervention implemented in eight primary schools located in low socioeconomic neighborhoods in the Netherlands, namely the KEIGAAF intervention. A qualitative, multiple-case study design was used to evaluate implementation and contextual factors affecting implementation. We used several qualitative data collection tools and applied inductive content analysis for coding the transcribed data. Codes were linked to the domains of the Consolidated Framework for Implementation Research. NVivo was used to support data analysis. The implementation process varied greatly across schools. This was due to the high level of bottom-up design of the intervention and differing contextual factors influencing implementation, such as differing starting situations. The mutual adaptation between top-down and bottom-up influences was a key element of the intervention. Feedback loops and the health promotion advisors played a crucial role by navigating between top-down and bottom-up. Implementing a mutual adaptation intervention is time-consuming but feasible.

## 1. Introduction

Given the significant amount of time children spend at school, the school environment has an important influence on children’s energy balance-related behaviors (EBRBs), i.e., sedentary behavior, physical activity (PA), and nutrition behavior. The World Health Organization advocates school health promotion [1]. However, not all school health promotion initiatives are successful. Systematic reviews and meta-analyses on the effectiveness of school-based physical activity (PA) and nutrition interventions on children’s energy balance-related behaviors and body mass index (BMI) found mainly mixed or inconclusive results [2,3,4,5,6,7,8,9,10,11]

To understand why interventions succeeded or failed, insight into what really happens during implementation is indispensable [12]. The number of studies regarding the implementation of school-based PA-promoting interventions is currently limited as compared with the number of effectiveness studies [13]. When school-based health-promoting intervention studies investigate implementation, it is often studied conventionally by assessing fidelity to the standardized intervention components [14]. This type of evaluation is appropriate for “one size fits all” evidence-based programs. 

However, “one size fits all” evidence-based interventions do not take into account contextual differences between settings. In contrast, interventions which allow local adaptation to ensure contextual fit do take these differences in context into account and are considered to be more appropriate, implementable, effective, and ultimately sustainable [15,16]. It is recommended that school health-promoting interventions should be sufficiently flexible to fit a specific context [14,15,16,17], and thus allow local adaptation. Mutual adaptation interventions are interventions in which adaptation of top-down principles and bottom-up development and implementation take place concurrently [16]. These interventions lead to different outputs and are implemented differently in different settings. To study the implementation of such an intervention and factors influencing implementation, a flexible evaluation approach and sensitivity regarding contextual influences and changes are required [14,18,19,20,21]. 

In this paper, we evaluated a mutual adaptation physical activity and nutrition intervention that was implemented in primary schools in the Netherlands [22], i.e., the KEIGAAF intervention. KEIGAAF is a Dutch acronym for “Chances in Eindhoven for a family-based approach by Fontys” (in Dutch, Kansen in EIndhoven voor GezinsAAnpak met Fontys) and refers to a local term for “super cool” [22]. We studied how KEIGAAF was implemented in primary schools and which contextual factors influenced implementation.

## 2. Materials and Methods 

### 2.1. Study Design

To study the implementation of KEIGAAF in the intervention schools, a qualitative, multiple-case study was conducted [23]. This process evaluation was part of a larger study, which also evaluated the effectiveness of KEIGAAF on children’s BMI z-score, physical activity, and nutrition behavior [22]. The process evaluation was conducted prior to the effectiveness study. 

Eight intervention schools were recruited in April and May 2016 in Eindhoven, a city in the south of the Netherlands. These schools were located in low socioeconomic neighborhoods. Eligibility criteria and recruitment strategies are described in detail in the study protocol [22]. 

### 2.2. The KEIGAAF Intervention

The KEIGAAF intervention was not a prepackaged program, but an approach that consisted of an interplay between top-down and bottom-up influences reinforcing each other in order to optimize the implementation of school-based PA and nutrition activities by ensuring contextual fit [22,24]. The overall aim was to create a school environment that stimulates children to be active and have healthy eating behaviors.

The top-down part of the KEIGAAF approach consisted of a steering committee of health behavioral experts and representatives of local organizations, who provided the basic principles of the intervention (see Figure 1) and supported the bottom-up part, for example, with scientific advice or financial resources. The bottom-up part consisted of local working groups that defined local intervention needs with respect to PA engagement and healthy nutrition and were responsible for the implementation of the intervention.

The working groups were encouraged to follow the steps proposed in the model by Van Kann et al. [24] which consisted of the following: (1) Compose a working group, (2) define local needs, (3) develop an activity plan, (4) apply for resources (additional ones), (5) implement PA and healthy nutrition-promoting activities, and (6) guarantee sustainability. Although these steps suggest a linear process, in reality it is a dynamic process with multiple feedback loops. The local working groups were supported by health-promoting (HP) advisors who advised the working groups on actions and effective activities. The HP advisors exchanged best practices and served as a link between the steering committee and the working groups. 

The working groups were advised on implementing a comprehensive and integrated set of PA and healthy nutrition-promoting activities. School health promotion is considered comprehensive and integrated when children’s health behaviors are promoted through health education, by the school’s physical and social environment, and beyond the school gates (thus also before and after school time) by engaging families and the wider community [1,25].

The KEIGAAF intervention started in April 2016 and lasted until June 2019 [22]. The intervention period consisted of a preparation period of about one year (April 2016 to April 2017) and an implementation period of two years (May 2017 to June 2018 and September 2018 to June 2019). When referring to year one, year two, and year three, we are referring to the preparation year (2016/2017), and the first (2017/2018) and second (2018/2019) year of implementation, respectively.

### 2.3. Study Setting and Study Population

The context of the intervention schools served as the study setting. By context, we mean “the set of circumstances or unique factors that surround a particular implementation effort” [26] (p. 52). To study contextual factors, we used the Consolidated Framework for Implementation Research (CFIR) [26], which is a tool to study factors that can influence intervention implementation [26,27]. It consists of five general domains which interact in complex ways to influence implementation effectiveness. These five domains are characteristics of the outer setting (the economic, political, and social context to which the organization belongs), the inner setting (attributes of structural, political, and cultural context), the individuals involved in the intervention (characteristics of the implementers), the intervention (divided into unadapted and adapted intervention), and the implementation process [26]. 

In this study, the outer setting was considered the external context of the school that could influence implementation, for example, national and local policies or collaborations with external organizations. The inner setting was considered the school environment, for example, involvement of school staff and the principal. The individuals involved in the implementation of the intervention were the working group members, i.e., schoolteachers, external professionals, parents, and the HP advisor.

In addition to the individuals involved in implementation, the study population consisted of all actors in the eight intervention schools, for example, schoolchildren, teachers and parents who were not involved in the working group but were part of the intervention setting. Observations focused on the entire context (i.e., outer and inner setting, and the entire study population). Interviews were conducted with a selection of the study population and members of the steering committee (described below). 

### 2.4. Data Collection

Data were collected in and around the schools. Data collection started in September 2016 and ended in June 2019 (Figure 2). For this, a flexible data collection approach was applied, i.e., data collection tools were added or removed during implementation to gain the best insights into the implementation process and the contextual influence. Ultimately, multiple qualitative evaluation methods were used to study implementation. The main researcher (SV-J) was involved in the implementation process as HP advisor. This engagement in practice enabled her to gain insight into the implementation process and sense the interplay between top-down and bottom-up influences, while supporting the implementation process [16]. Intuitive findings of the researcher concerning implementation were confirmed by the use of multiple qualitative measurement tools across multiple stakeholders. The Medical Ethics Committee of the Maastricht University Medical Centre approved the study (METC163027, national number: NL58554.068.16).

#### 2.4.1. Minutes of Working Group Meetings and Participatory Observations

From the start of the intervention until the end of the intervention, minutes were collected of each working group meeting (N = 113) and of the meetings of the steering committee (N = 8). Minutes of the working group meetings were mainly prepared by the HP advisor, and sometimes by another member of the working group (e.g., a teacher or a health professional). The minutes of these meetings provided in-depth information about the implementation plan ranging from informing the principals to developing and implementing plans and the (supporting) role of the steering committee in implementation.

After each working group meeting, the HP advisor made notes about their observations of the meeting (N = 89). Notes could concern the process (e.g., problems encountered by the members during implementation), interactions (e.g., communication between a working group member and a new partner), or other contextual influences (e.g., changes at the municipal level), but also included “soft” measurements, such as the atmosphere during meetings. These participatory observation notes were not shared with the working groups. In addition, a researcher regularly visited the participating schools to observe activities implemented there. Special attention was paid to contextual fit. These observations were recorded as notes. The participatory observation notes mainly served as secondary data to verify results found in the other data sources. They were used to give meaning and explanation to identified processes.

#### 2.4.2. School Scan

At the start of the intervention, the starting situation of the schools concerning the promotion of PA and healthy nutrition was assessed. At the beginning of the preparation year (school year 2016 to 2017), the school principal or a teacher filled in a school scan, which was an online questionnaire. The questionnaire was based on theoretical frameworks concerning the comprehensiveness of primary schools’ efforts towards PA and healthy nutrition promotion [1,25,28]. The school scan assessed the school’s physical education (PE), nutrition education, PA, and healthy nutrition-promoting policies; whether and how the physical school environment stimulated PA and healthy nutrition behavior; the involvement and support of staff; and parental involvement in the school’s PA and healthy nutrition promotion. At the beginning of each school year, the school scan was filled in online by the principal or the chair of the working group (i.e., a teacher). 

#### 2.4.3. Timeline Sessions

Initially, the working group members filled in a team climate checklist at the end of the preparation period to assess the climate for innovativeness within the working groups [22]. The team climate and changes within the team climate were expected to affect the implementation process. However, it did not appear feasible to measure follow-up due to a high participant turnover. Thus, this measure was considered inappropriate for this context. Instead, the narrative timeline technique was used at the end of the first and second year of implementation to evaluate activities implemented in that particular school year and to discuss perceived highlights and perceived failures during implementation with the working groups [29]. Participants concluded with a discussion about “what to do next”. The data were used to analyze the implementation process and factors influencing implementation. The HP advisor who assisted the working group acted as a participant in the timeline session. Another HP advisor moderated the session. The timeline sessions were recorded, and participants provided oral consent before the start of each session. In total, 16 timeline sessions were conducted, and 60 working group members participated in these sessions. The timeline sessions were conducted at school. Session duration ranged from 39 to 70 min. 

#### 2.4.4. Semi-Structured Interviews

In year three, a selective sample of principals (N = 5) and working group chairs (N = 4) was asked to participate in individual interviews to gain more insight into implementation at different levels (operational level and management level) and factors influencing implementation. This sample was chosen based on the diversity in implementation of the KEIGAAF approach. Additionally, members of the steering committee (N = 5) were asked to participate in individual interviews. All participants agreed to participate. The interviews were scheduled at the convenience of the participant and took place at the participant’s place of work. Before conducting the interview, a semi-structured interview guide was developed using the CFIR [26] (Appendix A). The interviews were recorded, and participants provided oral consent prior to the interview. The interview duration ranged from 28 to 57 min.

### 2.5. Data Coding and Analysis

Recorded data were transcribed verbatim. Inductive content analysis was performed when coding the data [30]. Two researchers (SV-J and AV) coded 10% of the transcribed interviews independently (one semi-structured interview and two timeline evaluations). These interviews were chosen at random. The two researchers discussed emerging themes and concepts, as well as constructed a preliminary codebook. After agreeing on the first version of the codebook, the first author (SV-J) continued the coding of the remaining interviews and adapted the codebook accordingly. Adaptations to the codebook were discussed with the second researcher. Subsequently, the codes were linked to the five domains of the CFIR [26]. Codes represented information concerning the implementation process (research Question 1), intervention factors, or the contextual factors, i.e., outer setting, inner setting, and characteristics of the individuals (research Question 2). Examples of codes linked to the process are “internal communication”, “modus operandi working group”, and “collaborations”. Examples of codes linked to the intervention are “KEIGAAF research” and “added value KEIGAAF’. Examples of codes linked to the context are “modus operandi school board” (outer setting), “school staff support” (inner setting), and “characteristics working group member” (individuals). Throughout the analysis, an iterative process was applied. The interpretation of the results was compared with the verbatim data. Data coding and analysis were supported by the use of NVivo 12.

## 3. Results

### 3.1. Implementation of the KEIGAAF Intervention

The timeline of the implementation of the KEIGAAF intervention is outlined in Figure 3. In general, the participating schools used similar basic processes concerning the implementation of the PA and nutrition-promoting activities, as they all followed most of the steps proposed by the model of Van Kann et al. [24]. All schools composed a working group (Step 1), defined local needs (Step 2), developed an activity plan (Step 3), and implemented PA and healthy nutrition-promoting activities (Step 5). All schools, except one, made use of the KEIGAAF budget to implement the activities (Step 4). Additionally, seven schools received financial resources from their school board to promote PA and nutrition at school. This school board was a member of the steering committee. The other school belonged to a different school board (not a member of the steering committee). Three schools also applied for and received financial resources from the national Health Promoting Schools committee during the intervention period. All schools, except two, guaranteed sustainability of their implemented activities (Step 6). 

#### 3.1.1. Formation of Working Group

At the beginning of the intervention period, the school principal and a physical education teacher was informed on the KEIGAAF intervention by the main researcher and the project leader. The school principal was instructed to form a working group. At the beginning, the working groups consisted mainly of teachers. Gradually, as the intervention developed, these teachers involved one, or more, parents in the working group. Four working groups did not succeed in involving a parent or involving a parent for a longer period of time. The working groups collaborated with one or more external professionals during the intervention period to implement activities. However, for four working groups, this collaboration with external professionals was minimal. Collaborations were primarily based on existing collaborations between school and an external organization. New collaborations between a school and external organizations were mainly initiated by the HP advisor or by the external organization after being informed about KEIGAAF by a member of the steering committee (e.g., a colleague). Principal involvement in the working group differed between schools. Principals were either directly involved as a member (two working groups), or more indirectly involved (four working groups), or the (main) principal was not involved at all (two working groups). Overall, the composition and size of the working groups differed per school (ranging from three to ten members) and the working groups were highly dynamic during the entire intervention period. Nevertheless, in almost all working groups there was at least one member (besides the HP advisor) who was involved during the whole intervention. Additionally, all working groups, except for one, were supported by the same HP advisor for the entire intervention period. 

#### 3.1.2. Development of Activity Plans

In the beginning, the HP advisor chaired the meetings of the working group and initiated the meetings. The HP advisor also ensured that needs were assessed at the beginning of the intervention period by brainstorming with the working group about needs and potential solutions. Additionally, the advisor discussed the results of the school scan and an environmental scan on the PA-friendliness of the school environment. The latter scan was conducted by the main researcher and a research assistant (more details about this scan can be found in the protocol paper [22]). The needs assessment provided the basis for the activity plans. The speed at which these plans were developed and the transition from the preparation phase to an implementation phase took place differed per working group. One working group was already in the implementation phase in the first school year, whereas the other working groups started implementation in the new school year (2017 to 2018). For four working groups, it took quite some time to consider and understand KEIGAAF as an intervention in which the working groups were responsible for developing and implementing activity plans rather than a predefined program of PA and nutrition activities. Additionally, working groups that considered their school’s existing PA promotion to be adequate took longer to develop an activity plan to promote PA. In general, the working group members were very practice oriented. As a result, the activity plans of the working groups were not very extensive and elaborated, and in most cases, short- and long-term aims of the activities and actions were not explicitly defined. However, three working groups did develop a more structured and deliberate activity plan for year three based on evaluations and experiences in the previous years. Although this practice-oriented thinking of the working groups did not facilitate plan development, it did facilitate implementation in four schools. 

#### 3.1.3. Implementation of Activity Plans

All working groups implemented PA and nutrition activities (Appendix A), but the degree and intensity differed. PA activities took less time to implement than nutrition activities because of their nature. Most PA activities were relatively simple activities or actions to implement (e.g., new PA material or supporting PA during recess) and were considered fun for the children, while nutrition activities were mainly policies and rules (e.g., not consuming sugar-sweetened beverages) that required support from multiple actors (school staff, parents, and children). In general, the implementation of the intervention was characterized by many feedback loops. Working groups went back and forth in the development and implementation of activity plans and modifying plans and activities based on evaluations and reactions of the target population. The HP advisor and research results supported this process. The HP advisor ensured that the basic intervention principles were met, advised working groups about evidence-based activities and best practices, and advised on how activities could be adapted to enhance contextual fit. To this end, every three months the HP advisors discussed the progress and implementation of the working groups and shared information. The HP advisor also played a key role in feeding back research results to the working groups (i.e., the data of the school scan and the environmental scan, and the results concerning the EBRBs of the children). At the same time, the HP advisor used the insights concerning the process (i.e., from participatory observations and from the timeline sessions) to modify their supporting strategies. For example, one school was not very active in implementation and maintained the brainstorm phase in year two. The HP advisor of this school anticipated this and took a more active role in developing the intervention plan and implementing it. As a result, the school implemented its plan in year three. Although this active involvement of the HP advisor facilitated implementation, it was not considered advantageous for intervention sustainability. The steering committee was informed on the process by two HP advisors. On the basis of these observations by the HP advisors, supporting strategies of the steering committee were adapted to local needs. For example, members of the steering committee informed colleagues on the intervention and requested them to collaborate with the schools in the working groups, and because of a less active implementation of the intervention, one member of the steering committee decided to motivate and support two working groups in implementing an active curriculum.

#### 3.1.4. Guarantee Sustainability

The working groups were advised to implement sustainable activities. In most cases, the HP advisor had to remind the working groups about this sustainable characteristic of activities and advised them to incorporate new practices into current practices. In addition to guaranteeing sustainability of the activities, the HP advisor also aimed to guarantee sustainability of the working group. Therefore, the HP advisor gradually handed over the role of chair to another working group member (a teacher). The readiness of a working group to continue independently of the HP advisor differed per school. For three working groups, the HP advisor was still the chair and initiator of the meetings at the end of the intervention period.

### 3.2. Factors Influencing the Implementation of the KEIGAAF Intervention

We refer to Figure 4 for a general overview of the factors per domain of the CFIR that facilitated or hindered implementation of the intervention.

#### 3.2.1. Intervention: Unadapted

##### Practical Support of HP Advisor and Financial Support

The practical support of the HP advisor was considered a facilitating factor in implementation by most working groups. Although some working groups were intrinsically motivated to implement the intervention, other working groups required the guidance and encouragement of the HP advisor. The HP advisor sought the best strategy to guide and encourage a working group. A good match between these strategies and the needs of the working group facilitated implementation. School health promotion was considered important by the working groups, but not a priority. Four working groups considered the KEIGAAF budget provided to them as a supporting factor in the implementation. 

##### Feedback Loops

The feedback loops in the intervention supported implementation. Three working groups experienced that the timeline sessions were very helpful in deciding on or improving their activity plans. In contrast to the usefulness of the timeline sessions, the results of the behavioral measurements of the children were demotivating for some schools because, in the short term, the children did not improve in their behavioral outcomes. One working group wanted to share good results with the parents to show the effectiveness of their efforts. However, they decided not to do so as no improvements in EBRBs were found at the first follow-up measurement. In general, the schools used the enthusiasm of the children (and in some cases the reactions of parents and teachers) as an indicator of intervention success.

##### Nutrition as Intervention Topic

Nutrition was considered a more difficult topic to address at school than physical activity. To promote healthy nutrition at school, support from multiple actors was needed. Especially parental support was considered important by the working groups, since parents provide the children’s snacks, lunch, drinks, and birthday treats. Support from school staff and principal support were also important for working group members to ensure that school nutrition policies and activities were implemented throughout school and in a consistent manner. However, parental and staff support was not self-evident in all schools, which inhibited implementation. At one school, not all parents supported the school’s policies concerning healthy nutrition, and thus gave the children sugar-sweetened beverages when they were expected to drink water. Some schools even felt unable to implement school nutrition policies because there was a lack of parental support. Staff support for nutrition policies and activities was lacking in three schools. In these schools, implementation of nutrition policies was delayed, nutrition activities were not implemented throughout the entire school, or there was no aim to promote healthy nutrition due to an expected lack of support. In schools where the principal perceived the children’s nutrition behavior to be the parents’ responsibility no nutrition policy was implemented. The principals of those schools did want to create awareness among the children about healthy nutrition, but this resulted in inconsistencies in healthy nutrition promotion by the teachers. In contrast, desirable nutritional changes among children were seen in the schools that implemented clear, formalized school nutrition policies.

#### 3.2.2. Outer Setting

##### National Health-Promoting Trends

In the Netherlands, schools are stimulated to become a healthy school [31]. To this end, a national committee awards a Health Promoting Schools (HPS) certificate to schools that meet the guidelines related to a particular HPS topic (e.g., PA or nutrition). In general, the guidelines of the HPS certificates state that the schools have to educate children about the health topic, identify health problems related to the topic, create a supportive social and physical school environment, and implement health policies [31]. The municipal health service is responsible for the local implementation of the HPS approach. In the intervention region, there were limited financial resources for the local public health sector to support schools in promoting health. A health promoter of the municipal health service advised school health coordinators on the requirements for the HPS certificates in group meetings. The guidelines of the national HPS committee concerning the HPS certificates enabled all schools to set priorities. Other national trends, such as the EU school fruit program [32] or national initiatives such as National Sports Week, also facilitated the schools in the implementation of PA and healthy nutrition promotion. 

##### Top-Down Influence of School Boards

The schools were governed by two different school boards, which were the two largest within the municipality. Seven intervention schools belonged to the same school board. Five years ago, this school board initiated a project to increase the children’s PA behavior by providing high-quality physical education. To this end, they employed qualified PE teachers at the schools. Additionally, the board financially supported schools in assigning a health coordinator within the school who was responsible for obtaining at least two HPS certificates. The school board’s demand to obtain the HPS certificates supported the KEIGAAF approach by accelerating the implementation of PA and nutrition-promoting activities. Six of the seven schools wanted to adhere to the school board’s demands concerning obtaining these certificates. As a result, these schools had obtained at least one HPS certificate at the end of the intervention period. The school that belonged to another school board did not feel this pressure to obtain one of the HPS certificates. They even believed that obtaining the HPS certificates was of no advantage to them. They were less active in the development of their activity plans and implementation. It took them longer to specify priorities.

##### Lack of Potential Partners

The intervention region is divided into seven districts. The participating schools were located in five different districts of the city. In the northern districts, there were different actors as compared with the southern district. Schools located in the northern districts perceived a lack of potential partners in health promotion in their neighborhood. Actors that were potential partners did not match the school’s working method. Even in regions with a potential partner, collaborations between school and potential partners were limited. As a result, most schools, implemented the activities only on the school premises, and the members of the school staff were the main implementer of the activities. In general, working group members lacked the capacity to form these collaborations or did not see the necessity to do so. Three schools did succeed in working together with local partners in the implementation of their intervention. This was because the principal, a parent, or the HP advisor initiated this collaboration. A lack of potential partners or limited collaborations between school and potential partners hindered the implementation of comprehensive PA and healthy nutrition promotion.

#### 3.2.3. Inner Setting (Schools)

##### Starting Situation of Schools

The defined needs and the activity plans developed by the working groups were dependent on the school’s starting situation concerning PA and healthy nutrition promotion. A good starting situation for implementation was a situation in which the working group considered that the current situation had to change. This was mainly the case when there was limited PA or healthy nutrition promotion at school, while the working group perceived that the children needed to improve their EBRBs. When the working group considered that their school was already making much effort regarding PA and nutrition promotion, implementation was hindered. Hardly any of the schools focused on their own current practices concerning PA promotion with the aim to improve these practices; they were considered normal and consequently overlooked as a potential unit of change.

##### Low Parental Support

All schools expressed that a lack of parental involvement in school activities (whether health-related or not) was a common problem. This low parental involvement was also evident in the number of parents participating in the working group; only four schools had at least one parent participating in the working group during the whole intervention period. Schools were hesitant to ask parents to participate in the working groups. 

##### Support of School Staff and Principal

School staff support facilitated the integration of activities and policies within the school. Some working group members motivated the school staff to support implementation. Creating school staff support was easier when the number of teachers in the working group was high relative to the total number of school staff. The support of the principal was also a facilitator in the implementation of the intervention. All initial principals decided to participate in the intervention. However, there was a high turnover in principals at most schools. In these schools, the new principal had not decided upon participation. This did not necessarily mean that implementation was hindered in these schools. Most of these new principals were very supportive of the intervention. The principals of the seven schools of the same school board were all instructed to obtain the HPS certificate by the school board, and most of them wanted to adhere to that obligation. The schools that were most active in implementation had a principal who supported the working group (e.g., by providing hours), agreed on decisions made by the working group, and demanded that the rest of the school staff support the implementation of this plan. To promote the feeling of ownership of KEIGAAF by the school staff, it was best if the principal was autonomy supportive instead of controlling. 

##### Employee Turnover

Besides a high turnover in school principals, all schools experienced employee turnover during the intervention period. This also resulted in working group members being replaced, added, or removed. At some schools, this employee turnover inhibited implementation because of poor communication between the leaving employee(s) and the new employee(s) or uncertainty about division of tasks. However, in most schools, the employee turnover facilitated the implementation of the intervention, because the new members were more proactive, had more decisional power, or because the changes could be more easily implemented given the new teachers’ unfamiliarity with the old practices.

#### 3.2.4. Individuals (Working Group)

##### Misinterpretation of the Intervention 

Misinterpretation of the intervention approach and the intervention objectives inhibited implementation. For example, one working group did not develop their own intervention plan, but waited for activities to be delivered and implemented by external organizations, and two working groups thought that they were expected to implement new things, while an objective could also be to bring about more coherence into the existing school PA and healthy nutrition promotion. 

##### Practice-Oriented Thinking

The practice-oriented thinking of the working groups was considered inherent to the way of working in schools. This attribute of schools did not facilitate the development of a deliberated and sustainable plan, but it did facilitate implementation in most schools. 

##### Champion

A factor facilitating the integration of the activities within a school was the presence of a champion. This champion was characterized by enthusiasm, felt a great need to improve the children’s health behavior, quickly switched from a thinking mode to an acting mode, and focused on opportunities instead of obstacles. Primarily, the champion informed and involved other school staff. Three schools had clear champions. This was a teacher and the chair of the working group. Working groups that did not have a champion and did not feel an intrinsic need to improve the children’s health behavior but were more externally steered (e.g., requirement of the school board) to participate in the KEIGAAF working groups were less active throughout the intervention period. 

##### Positive Dynamics 

Good interaction between working group members, including constructive communication between the working group members and the HP advisor, facilitated the process of developing and implementing. Working groups that were characterized by enthusiastic, proactive members who felt a need to change things and invested time in this, collaborated well, and divided implementation tasks had more output as compared with working groups that were less enthusiastic or proactive. 

#### 3.2.5. Intervention: Adapted

##### Adaptation

The KEIGAAF intervention was characterized by a high degree of adaptation and local tailoring. This facilitated the implementation of activities that were suitable for the local context. It also enhanced feelings of ownership and sustainability of the bottom-up approach (i.e., the working groups). At the end of the intervention period, six schools decided to continue with KEIGAAF. Although the content of the intervention was very flexible, there was no flexibility in the application of working groups. The working groups were considered one of the main elements of the mutual adaptation intervention. The practice of working groups worked well for five schools but was less suitable for three schools. These schools preferred a more top-down approach. For example, one school required external organizations to offer an intervention, while the other school made all the decisions for the school’s PA promotion via the principal, who did not want to be actively involved in the working group. As a result, these working groups hardly developed an activity plan (if one was developed at all). 

## 4. Discussion

We studied how a mutual adaptation intervention aimed at promoting children’s physical activity and nutrition behavior was implemented in eight primary schools located in low socioeconomic neighborhoods. Although the eight schools were located in the same municipality, the schools differed greatly in the implementation of PA and healthy nutrition-promoting activities. This was caused by the high level of bottom-up design of the intervention and differences in the contexts of the schools. Secondly, we studied which contextual factors influenced implementation. Schools had differing starting situations concerning PA and healthy nutrition promotion andthey differed in received parental support, staff and principal support, and in employee turnover. Moreover, differences in characteristics related to the working group members, i.e., interpretation of the approach, the degree of practice-oriented thinking, the presence of one or more champions, and the dynamics within the working group, but also the differing degrees of influence of factors within the outer setting (i.e., support from the school board, national health-promoting trends, and the presence of and the capacity to collaborate with potential partners) resulted in different implementation processes. Other studies also found these contextual factors to facilitate and hinder implementation of school health-promoting interventions [33,34,35,36,37,38,39,40,41].

The implementation of the KEIGAAF intervention was the result of an interplay between top-down influences and bottom-up development and implementation which led to adaptation to the local context [16]. This mutual adaptation was a key element of KEIGAAF, with the feedback loops and the HP advisors playing a crucial role in this adaptation. For example, feedback on the schools’ physical and social health-promoting environment was used to define local needs and behavioral outcome measures to catalyze implementation processes. However, the latter appeared inappropriate for the schools. Reporting short-term, null effects of the first interventions on behavioral outcomes was considered demotivating by the working groups. Instead, children’s enjoyment was a preferred indicator for implementation success by the working groups. Therefore, it would have been better to evaluate and report back on children’s enjoyment and preferences [42]. To support working groups in implementing context-appropriate activities, the advisor had to find the right balance between top-down and bottom-up influences for each school. This meant that for the one working group, the HP advisor had to take a more active role in implementation, while for the other working group merely informative support was appropriate. For example, a working group that experienced difficulties with the implementation of the intervention outside of school hours and the school premises due to a lack of partners or inability to find potential partners required support from the HP advisor to find, inform, and connect partners to the working group. Whereas another working group already had a strong external network and required a different type of support, for example, assistance in obtaining parental support. In that case, the HP advisor supported finding the right strategies to involve parents. Thus, the HP advisor also played an important role in overcoming barriers experienced during implementation. However, some barriers were difficult to overcome, for example, working groups that continued to misinterpret the KEIGAAF approach. In such a case, the HP advisor eventually decided to decrease support because it was not the right time for the working group and school to implement the intervention. Since this balance between top-down influences and bottom-up design differed per school and throughout the process, the advisor had to have adaptive management skills and be context sensitive. Being context sensitive meant having all senses open and observing physical structures and organizational dynamics [43] prior to and during intervention implementation. To achieve this, engagement in practice was essential [16].

Interestingly, the need for mutual adaptation between top-down intervention principles and bottom-up changes in school-based health promotion had already been acknowledged in 1976 [44]. However, applying an adaptive intervention instead of implementing a predefined set of activities is still not common practice in school-based interventions. This can be explained by a certain degree of “lack of control” as well as fear of “cherry picking” (e.g., implementing convenient intervention elements) affecting the key components of an intervention. However, intervention adaptation (i.e., local tailoring) is not a threat to intervention effectiveness when intervention functionality is maintained [17]. Adaptation is even considered necessary to maximize the effects [17].

There is currently no guidance on how to adapt evidence-based interventions for new contexts [45], but we know that the ability to mutually adapt top-down effective principles and bottom-up changes “is a special niche and it’s not easy to do” [16] (p. 179). It takes skills, perseverance, and time. Informing the schools about an intervention with an unknown output was a challenging process which had to be repeated multiple times throughout implementation. The HP advisor had to familiarize themself with the school and had to gain the trust of the working group [33]. Both investments were essential, but time-consuming. This is outweighed by the high potential of sustainability of the output of the intervention because of a high perceived ownership by the schools due to contextual fit [16,36]. Important lessons we have learned from implementing the KEIGAAF intervention are listed in Figure 5.

### Strengths and Limitations of the Study

This is one of the few studies examining the implementation of school-based PA and nutrition-promoting activities with a focus on context [46]. The combination of the use of different evaluation tools, such as the timeline sessions and the school scan, and our continuous presence in practice provided us with a deep understanding of this context. The engagement of the researcher in practice was a strength of this study but can also be seen as a limitation because of a loss of objectivity. To increase the objectivity of the interpretation of the results, a second researcher was involved in the data coding and analysis. The flexibility in the research approach was another strength of the study [19]. For example, initially the team climate in the working groups (e.g., a safe environment to share ideas) was measured using validated questionnaires at the end of the preparation year with the intention of repeating this measurement at the end of years two and three, because this outcome was expected to impact the process. However, follow-up was impossible due to a high participant turnover. Thus, this measure was not repeated. Additionally, more insight into the process was needed and the minutes of the meetings and observations did not provide enough detail, thus the timeline sessions were used. 

There are also some limitations to this study that should be mentioned. First, it was not possible to fully assess the implementation process (e.g., full insight into the integration of the activities in school or the spill-over effects of activities from school to home) or to study every contextual factor (i.e., contextual factors broader than the school context) due to time and resource limitations. In addition, we aimed to minimize participant burden [19]. Second, the results regarding the process and contextual factors are specific for this intervention region and are not generalizable to other regions. This is an inherent aspect of the KEIGAAF intervention. When implementing the KEIGAAF intervention in other regions, it is up to the researchers and HP advisors to gain insight into the contextual factors of that region that potentially influence implementation. A final limitation of the study was that the duration was relatively short to be able to measure sustainability. The KEIGAAF approach was intended to be actively developed and implemented in schools with the support of the HP advisor in years one and two. In year three, the support of the HP advisor was intended to diminish. That last year would then give us a first insight into the potential sustainability. However, this was not realistic because implementation was a continuous process of trial-and-error, and integration and adjustment of plans, activities, and actions required more time than the three years of evaluation. Only preliminary insights into the sustainability of the output of the approach could be acquired. Further research is needed concerning the sustainability of the output of the KEIGAAF intervention, especially when a context requires a more top-down approach. 

## 5. Conclusions

This study showed that a mutual adaptation physical activity and nutrition intervention was implemented differently in eight primary schools located in low socioeconomic neighborhoods in the same municipality due to the high level of bottom-up design of the intervention and the differences in influence of facilitating and hindering contextual factors. Important lessons were learned from implementation. One of these lessons was that implementation of this mutual adaptation intervention is time-consuming but feasible. Adaptation of key intervention principles to the school context and current practices is essential. Health promotion advisors play a crucial role in this adaptation by navigating the middle path between top-down and bottom-up influences in a particular school context, as well as feedback loops between other supporting activities and implementation.

## Figures and Tables

**Figure 1 ijerph-17-00751-f001:**
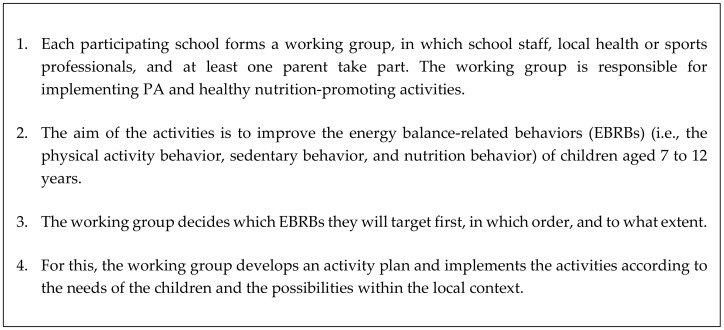
Basic intervention principles as provided by the steering committee.

**Figure 2 ijerph-17-00751-f002:**
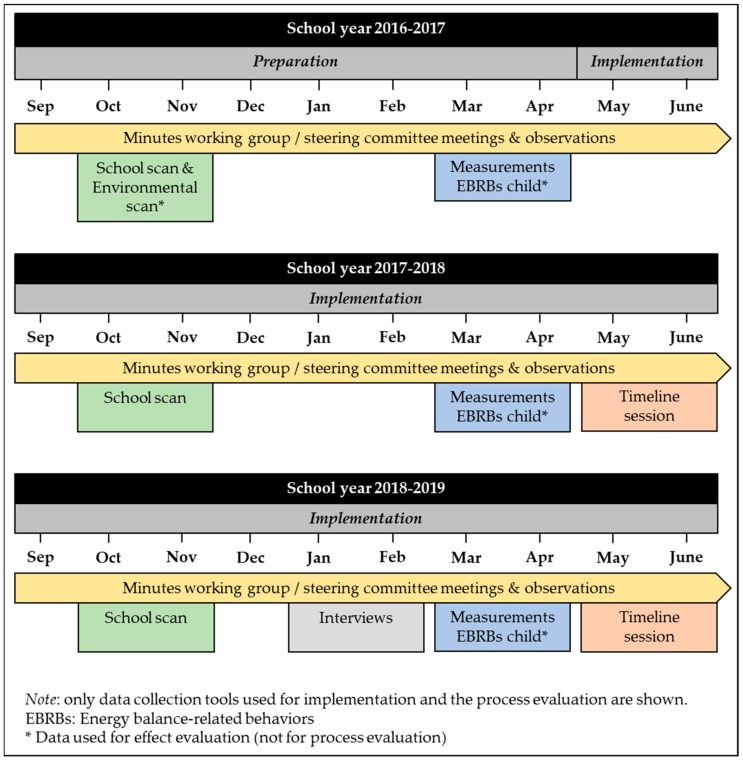
Timeline of data collection for process evaluation.

**Figure 3 ijerph-17-00751-f003:**
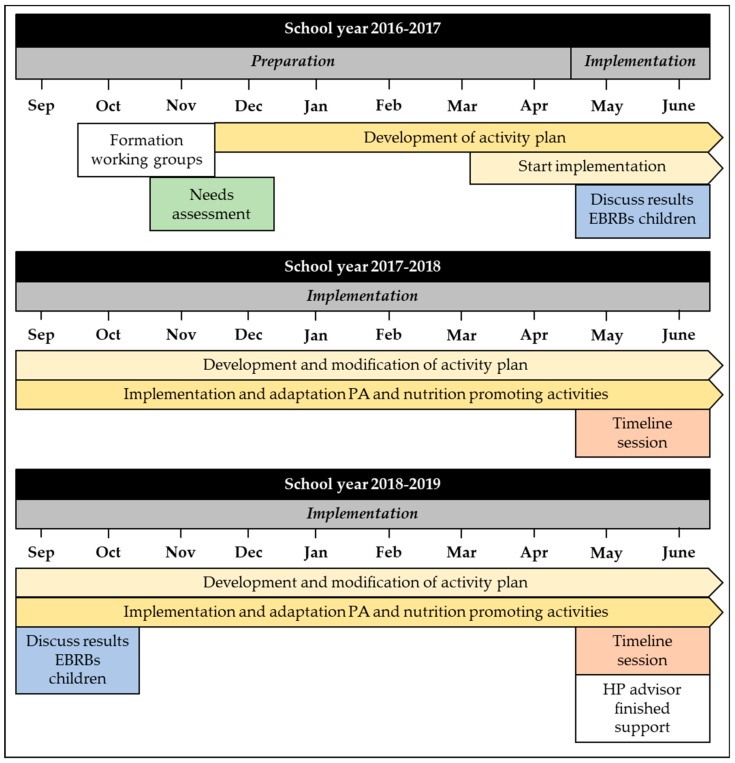
Timeline of the implementation process of the KEIGAAF intervention.

**Figure 4 ijerph-17-00751-f004:**
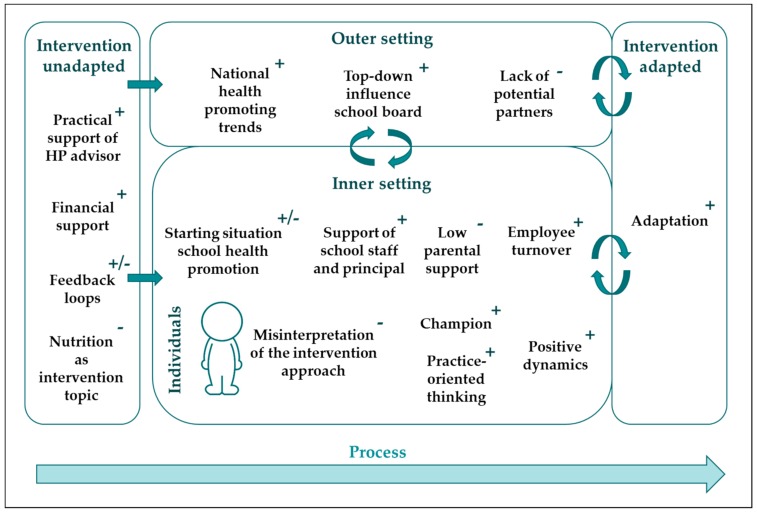
Facilitating (+) and hindering (-) factors in the implementation of PA and healthy nutrition-promoting activities in the schools per domain of the Consolidated Framework for Implementation Research (CFIR) (adapted from [26]).

**Figure 5 ijerph-17-00751-f005:**
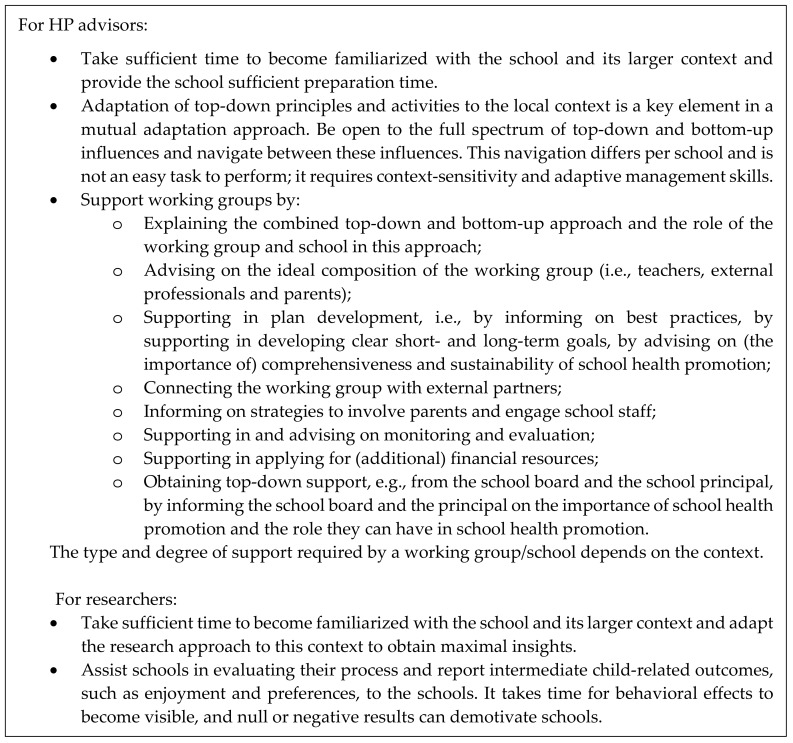
Important lessons learned from implementing the KEIGAAF intervention.

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
