# Peer review of "Implementation of KEIGAAF in Primary Schools: A Mutual Adaptation Physical Activity and Nutrition Intervention"

_ijerph, 2020, doi:10.3390/ijerph17030751_

Round 1

Reviewer 1 Report

Thank you for the opportunity to review the paper, “Process evaluation of KEIGAAF: a mutual adaptation approach to the implementation of school-based physical activity and nutrition activities”. This manuscript describes the implementation and evaluation of a school-based health promotion program in the Netherlands. There are a number of issues with the presentation of the information in this paper that dampen enthusiasm. Please see comments below:

Major Comments:

The authors should consider reorganizing the paper to ensure there is no overlap with previously published information. For example, Figure 1 has already been published as part of the protocol paper. Further, there are many aspects of this paper that are not aligned with the protocol paper (e.g., in this paper the authors state that schools signed up for participation in Oct and Nov 2016 but this is not what is said in the protocol paper). This is somewhat inconsistent with what is presented in Figure 2. How were schools and working group members recruited? There is often not enough detail about important aspects (e.g., methods and design of the evaluation, including elaboration of the types of evaluation methods used). The protocol paper is organized in a more reader-friendly manner and flows nicely. What does KEIGAAF stand for? This acronym is not mentioned in the paper. The introduction section is missing information on why school-based health promotion programs are needed-why are they valuable? What would be helpful is a timeline similar to what is presented in the protocol paper but one that shows the implementation and evaluation components, associated evaluation tools, etc. It is not adequate to say, line 126 “numerous participants”. Who participated in which method and at what time points? How were schools and working group members recruited? The tables and figures have duplicate text and could be substantially condensed. Please review all figures and tables for readability and clarity. For example, for all schools, the text reads, “x or y were informed of the approach” and “the HP advisor was chair”. This is not needed in the table. As written, these tables are not user friendly and are repetitive. Budget is mentioned as a barrier in the text and but not in Figure 3. There is one sentence on how the actual results of the intervention were barriers to implementation. How did the authors adapt the process based on these results? At KEIGAFF? Within each school?

Author Response

Dear reviewer,

Thank you for taking the time to review our manuscript. The valuable comments enabled us to improve our manuscript substantially, for which we are very grateful. We took each comment into account and made the following changes in our manuscript (highlighted in red in the manuscript):

- We ensured a better alignment in terminology and information presented in the protocol paper. For example, in the title we changed ‘KEIGAAF approach to ‘KEIGAAF intervention’. Contrary, we now used the term ‘a mutual adaptation intervention’ instead of ‘context-based intervention’. The use of this terminology evolved over time and better reflects the process of implementation of KEIGAAF .

- We added some information on the importance of school health promotion in the introduction. However, we consider this information to be well-known and therefore, we decided not to spend a large amount of text on this and maintained to focus on the importance of conducting a process evaluation (lines 49-59) and the importance of implementing mutual adaptation interventions (lines 53-63).

- We added the acronym of KEIGAAF in the introduction (a Dutch acronym for ‘Changes in Eindhoven for a family-based approach by Fontys’) and some explanation on this acronym (lines 62-64).

- We agree with both reviewers that the structure in the ‘Materials and Methods’ section in the original manuscript does not enhance readability of the manuscript. Therefore, we restructured the ‘Materials and Methods’-section. For this, we used the structure of the protocol paper which was considered more reader-friendly.

- Since we are referring to our protocol paper, we decided to delete Figure 1 (visual presentation of the mutual adaptation approach). This figure is already presented in the protocol paper.

- We decided to incorporate the information on the Consolidated Framework for Implementation Research (CFIR) of Damschroder and colleagues (2009) in the section ‘Study setting and study population’ (lines 110-129) instead of in the introduction to explain what we meant with context (for this we used the definition of Damschroder et al., 2009) and that we divided the context into different settings (outer setting, inner setting and individuals) based on this framework. To explain these settings, we gave some examples (lines 120-124).

- To guide the reader through the ‘Data collection’ section, we added a figure of a timeline of data collection (Figure 1, line 143) and presented the data collection tools in chronological order starting with the minutes of the working group and steering committee meetings and participatory observations (lines 145-162), followed by the school scan (lines 163-173), timeline sessions (174-188) and semi-structured interviews (189-199). The lay-out of this figure resembled the timeline-figure used in the protocol paper, which was considered more reader-friendly.

- We ensured that the information presented in the protocol paper aligned the information presented in ‘Materials and Methods’ section of this paper. For this, we thoroughly compared both papers again and ensured that the content was aligned. For example, while we mentioned in the initial manuscript that the schools participated in October and November 2016, schools were already recruited in April 2016. In this revised version, we now clearly stated that the intervention started in April 2016 and lasted until June 2019 (line 100). We apologize for this inconsistency.

- In the Data coding and analysis part we added examples of the inductive codes related to the different domains of the CFIR to give an idea of which codes were linked to the domains of the CFIR (lines 211-215).

- We decided to replace Figure 2 (representing a timeline on the implementation and evaluation of the intervention) by a more reader-friendly figure of a timeline of intervention implementation (line 233), which resembled the lay-out of the timeline used in Figure 1 (and the timeline used in the protocol paper), which is considered to be more reader-friendly . Additionally, instead of explaining intervention implementation in a table, we decided to provide a textual explanation of the implementation of the intervention and mentioned the main differences between schools instead of providing details per school (lines 235-313). Thus, we deleted Table 1 on the development and implementation of physical activity (PA) and healthy nutrition promoting activities in the eight schools, because the information was too specific and not reader-friendly.

- For the same reasons as mentioned for the deletion of Table 1, we also decided to delete Table 2 on the facilitating and hindering factors in the implementation of PA and healthy nutrition promoting activities in the schools. By doing this, we placed more emphasis on Figure 3, in which facilitating and hindering factors are placed within the domains of the CFIR (line 319). We ensured that the information provided in this figure is consistent with the information provided in text.

- We agree with the reviewer that the description of the implementation process could be improved. We optimized the description and had a good look at the factors presented within the different domains of the CFIR and compared these factors with the factors presented by Damschroder et al. (2009). We concluded that intervention elements like ‘Practical support of HP advisor’, ‘Financial support’, ‘Feedback loops’ and ‘Nutrition as intervention topic’ could be better interpreted by readers when considered as ‘the unadapted intervention-domain’, whereas only ‘Adaptation’ belonged to the adapted intervention-domain. Besides, the ‘Starting situation of school’ concerning PA and healthy nutrition promotion was considered a factor belonging to the Inner Setting-domain and not the Intervention-domain. We improved the content of Figure 3 (line 319) and explained these factors in text (Practical support of HP advisor and financial support’, lines 325-332; ‘Feedback loops, lines 334-343, ‘Nutrition as intervention topic’, lines 346-363; ‘Starting situation of school’, lines 409-419; and ‘Adaptation’, lines 485-497).

- In Box 2, we presented the important lessons learned from the implementation of the KEIGAAF intervention. Recommendations are aimed at HP advisors and researchers. To make it more clear, which recommendations are aimed at HP advisors and which at researchers, we made a distinction between recommendations for HP advisors and recommendations for researchers (line 549).

- To provide more punctual answers to the main research questions, we slightly changed a part of the conclusion (lines 585-594). Changes are highlighted in red.

- Instead of adding the starting situation of the schools concerning PA and healthy nutrition promotion as supplementary material, we decided to supplement a table on the output of the schools (i.e., PA and nutrition-promoting activities implemented in the schools). The activities that were implemented or enhanced during the intervention period were shown in Italics.

Reference:

Damschroder, L.J.; Aron, D.C.; Keith, R.E.; Kirsh, S.R.; Alexander, J.A.; Lowery, J.C. Fostering implementation of health services research findings into practice: a consolidated framework for advancing implementation science. Implement Sci 2009, 4, 50, doi:10.1186/1748-5908-4-50.

Reviewer 2 Report

Overall comments: the manuscript addresses an important area with a structured approach. The most consistent issues that I had reading the article were that, 1) the manuscript is particularly articulate to read, therefore is not so clear; 2) the manuscript is strictly linked to the previous one published last year presenting the protocol. Therefore it would be more appropriate to be consistent also in the terminology; 3) reading the abstract the main aims do not coincide with the research questions presented in the manuscript; the main research questions do not receive relevant answers in the results sections and in the respective order. Tables are really difficult to read and not add relevant information. Major revisions are needed.

Lines 2-4 The title is confounding and not immediate. Reading the study protocol you may be more consistent with it.

Lines 20-22 It is not clear if your aim was to compare the schools or to evaluate the KEIGAAF process.

Lines 31-32 Authors, with this sentence editorialize too much and make assumptions that aren’t supported by their research.

Lines 56-58 The two sentences are redundant.

Lines 59-60 Is the CFIR a tool to evaluate the context?

Lines 68-70 This two research questions do not correspond  to the ones presented in the abstract.

Lines 72-73 Could be combined.

Lines 74-78 You could first explain what is the KEIGAAF approach and then the purpose.

Line 78 The figure is presented in the previous manuscript, in this case it diminishes the readability of the article. More details about the KEIGAFF approach as the acronym spelling may improve fruition.

Lines 92-93 This is awkward wording.

Line 93 It is not clear what do you mean for  evidence-based activities.

Lines 94-95 This statement will be more appropriate in the next section.

Lines 100-104 These statement are inappropriate in the method section.

Lines 113-120 This paragraph is not so clear and it seem s to be more pertinent with the section 2.4.

Lines 123-128 To make the reading smoother this section can be combined with the previous one.

Lines 126-128 This is not an explanation of the setting and population.

Line 129 Overall the data collection section is difficult to understand since the generic explanation of the process is not consistent with these passages that now suddenly appear.

Line 130 Why are you starting from the timeline session if they are the last action you did?

Lines 131-132 Repetitions are present.

Lines 133-134 it seems a result not a method.

Lines 135-136 What it means?

Line 140 This is a result.

Line 141 Why were these interviews conducted? Who selected this purposive sample and on which criterion? How was developed the semi-structured interview guide?

Line 150 This section is not clear at all. Why the minutes of working groups meeting are important? Furthermore, what are the working group meeting?

Line 157 It is 2.4.4. and what do you mean for participatory observations?

Line 159 The number of notes is a result and why is it relevant?

Lines 163-167 This is awkward wording.

Lines 168 Why the starting situation is mentioned at the end of methods section?

Lines 185 It is not clear how the codes were linked to the five domains of the CFIR.

Figure 2. It is clear, only December 2016 is not readable. The problem is that the manuscript is not in line with the figure. The distinction between implementation and research is not well addressed.

Table 1. The table as currently worded is difficult to read and it does not help understanding differences between schools. You need to represent only the main different points.

Line 212 The remaining domain because the  process domain was explained with table 1? The process domain was the last one. Be more consistent with  adapted and unadapted?

Figure 3. It is a summary of the facilitating  and hindering factors in the implementation of the KEIGAAF per domain of the CFIR. The following table is redundant with the sections developed below and did not add useful information. This scheme can be a good guide to developing the points in the manuscript but in a consistent manner (see “Wrong Interpretation of the KEIGAAF approach” in the figure and  “Interpretation of the KEIGAAF approach” in the text).

Line 247-266 Be more concise with the main differences between schools.

Lines 297-298 You need to show only the results. You need  to avoid comments in the whole results section.

Lines 398-404 This section is redundant.

Line 447 It is not clear how the reference could help in the summary of the results.

Box 2. HP advisors and researches are used sometimes interchangeably.

Lines 496-497 these tools need to be clearly described in the methods section and introduced better at the beginning.

Lines 499-500 This passage is not clear in the manuscript.

 Lines 501-504 It is not sufficiently clear.

Lines 523-530 Conclusion did not give punctual answers to the main aims.

References are numbered twice.

Check reference 18.

Where are reported the results of the questions showed in the supplemental file 1?

Supplemental file 2. The physical education duration row you do not need to repeat PE for each column.

Author Response

Dear reviewer,

Thank you for taking the time to review our manuscript and the positive remark. To improve reader-friendliness of the manuscript, we restructured the ‘Methods and Materials’ sections and deleted the two tables on the implementation of the intervention within the schools and the factors facilitating and hindering implementation within the schools. Instead, we aimed to give a more general overview and highlight differences between schools without focusing on specific schools in text. Secondly, we followed the suggestion to better align terminology used in the protocol paper and in this manuscript.

The valuable comments enabled us to improve our manuscript substantially, for which we are very grateful. We took each comment into account and made the following changes in our manuscript (highlighted in red in the manuscript):

- We ensured a better alignment in terminology and information presented in the protocol paper. For example, in the title we changed ‘KEIGAAF approach to ‘KEIGAAF intervention’. Contrary, we now used the term ‘a mutual adaptation intervention’ instead of ‘context-based intervention’. The use of this terminology evolved over time and better reflects the process of implementation of KEIGAAF .

- We added some information on the importance of school health promotion in the introduction. However, we consider this information to be well-known and therefore, we decided not to spend a large amount of text on this and maintained to focus on the importance of conducting a process evaluation (lines 49-59) and the importance of implementing mutual adaptation interventions (lines 53-63).

- We added the acronym of KEIGAAF in the introduction (a Dutch acronym for ‘Changes in Eindhoven for a family-based approach by Fontys’) and some explanation on this acronym (lines 62-64).

- We agree with both reviewers that the structure in the ‘Materials and Methods’ section in the original manuscript does not enhance readability of the manuscript. Therefore, we restructured the ‘Materials and Methods’-section. For this, we used the structure of the protocol paper which was considered more reader-friendly.

- Since we are referring to our protocol paper, we decided to delete Figure 1 (visual presentation of the mutual adaptation approach). This figure is already presented in the protocol paper.

- We decided to incorporate the information on the Consolidated Framework for Implementation Research (CFIR) of Damschroder and colleagues (2009) in the section ‘Study setting and study population’ (lines 110-129) instead of in the introduction to explain what we meant with context (for this we used the definition of Damschroder et al., 2009) and that we divided the context into different settings (outer setting, inner setting and individuals) based on this framework. To explain these settings, we gave some examples (lines 120-124).

- To guide the reader through the ‘Data collection’ section, we added a figure of a timeline of data collection (Figure 1, line 143) and presented the data collection tools in chronological order starting with the minutes of the working group and steering committee meetings and participatory observations (lines 145-162), followed by the school scan (lines 163-173), timeline sessions (174-188) and semi-structured interviews (189-199). The lay-out of this figure resembled the timeline-figure used in the protocol paper, which was considered more reader-friendly.

- We ensured that the information presented in the protocol paper aligned the information presented in ‘Materials and Methods’ section of this paper. For this, we thoroughly compared both papers again and ensured that the content was aligned. For example, while we mentioned in the initial manuscript that the schools participated in October and November 2016, schools were already recruited in April 2016. In this revised version, we now clearly stated that the intervention started in April 2016 and lasted until June 2019 (line 100). We apologize for this inconsistency.

- In the Data coding and analysis part we added examples of the inductive codes related to the different domains of the CFIR to give an idea of which codes were linked to the domains of the CFIR (lines 211-215).

- We decided to replace Figure 2 (representing a timeline on the implementation and evaluation of the intervention) by a more reader-friendly figure of a timeline of intervention implementation (line 233), which resembled the lay-out of the timeline used in Figure 1 (and the timeline used in the protocol paper), which is considered to be more reader-friendly . Additionally, instead of explaining intervention implementation in a table, we decided to provide a textual explanation of the implementation of the intervention and mentioned the main differences between schools instead of providing details per school (lines 235-313). Thus, we deleted Table 1 on the development and implementation of physical activity (PA) and healthy nutrition promoting activities in the eight schools, because the information was too specific and not reader-friendly.

- For the same reasons as mentioned for the deletion of Table 1, we also decided to delete Table 2 on the facilitating and hindering factors in the implementation of PA and healthy nutrition promoting activities in the schools. By doing this, we placed more emphasis on Figure 3, in which facilitating and hindering factors are placed within the domains of the CFIR (line 319). We ensured that the information provided in this figure is consistent with the information provided in text.

- We agree with the reviewer that the description of the implementation process could be improved. We optimized the description and had a good look at the factors presented within the different domains of the CFIR and compared these factors with the factors presented by Damschroder et al. (2009). We concluded that intervention elements like ‘Practical support of HP advisor’, ‘Financial support’, ‘Feedback loops’ and ‘Nutrition as intervention topic’ could be better interpreted by readers when considered as ‘the unadapted intervention-domain’, whereas only ‘Adaptation’ belonged to the adapted intervention-domain. Besides, the ‘Starting situation of school’ concerning PA and healthy nutrition promotion was considered a factor belonging to the Inner Setting-domain and not the Intervention-domain. We improved the content of Figure 3 (line 319) and explained these factors in text (Practical support of HP advisor and financial support’, lines 325-332; ‘Feedback loops, lines 334-343, ‘Nutrition as intervention topic’, lines 346-363; ‘Starting situation of school’, lines 409-419; and ‘Adaptation’, lines 485-497).

- In Box 2, we presented the important lessons learned from the implementation of the KEIGAAF intervention. Recommendations are aimed at HP advisors and researchers. To make it more clear, which recommendations are aimed at HP advisors and which at researchers, we made a distinction between recommendations for HP advisors and recommendations for researchers (line 549).

- To provide more punctual answers to the main research questions, we slightly changed a part of the conclusion (lines 585-594). Changes are highlighted in red.

- Instead of adding the starting situation of the schools concerning PA and healthy nutrition promotion as supplementary material, we decided to supplement a table on the output of the schools (i.e., PA and nutrition-promoting activities implemented in the schools). The activities that were implemented or enhanced during the intervention period were shown in Italics.

Minor comments reviewer and response to each comment point-by-point:

Lines 2-4 The title is confounding and not immediate. Reading the study protocol you may be more consistent with it.

Thank you for this suggestion. We partly changed the title to ensure alignment with the protocol paper (line 1), i.e. intervention instead of approach. The adding of the concept mutual adaptation evolved throughout the process and we kept this term in the title as it reflects the process of the implementation best.

Lines 20-22 It is not clear if your aim was to compare the schools or to evaluate the KEIGAAF process.

We understand that this sentence is perhaps slightly confusing; therefore, we changed it into ‘A qualitative, multiple-case study design was used to study implementation and contextual factors affecting implementation’ (lined 22-23). We considered each schools as a unique case, but we presented the results of the overall implementation and highlighted differences between schools.

Lines 31-32 Authors, with this sentence editorialize too much and make assumptions that aren’t supported by their research.

We agree that the claim on sustainable activities is partly based on theoretical assumptions.. We removed this sentence from the abstract and the conclusion.

Lines 56-58 The two sentences are redundant.

We consider these sentences a rationale for this paper. We now integrated these sentences in the aim of the study.

Lines 59-60 Is the CFIR a tool to evaluate the context? /

The CFIR is a tool to evaluate factors that may influence implementation. We changed this sentence (line 115) and replaced this information to the methods-section (lines 114-121).

Lines 68-70 This two research questions do not correspond to the ones presented in the abstract.

We apologize for this inconsistency and now ensured that the research questions presented in the abstract aligned the research questions in the introduction (lines 22-23 and lines 67-68).

Lines 72-73 Could be combined.

We agree with that this does not enhance readability of the manuscript. Therefore, we combined the two headings into one heading.

Lines 74-78 You could first explain what is the KEIGAAF approach and then the purpose.

As suggested, we now first explained what KEIGAAF is and then the overall aim (lines 79-83).

Line 78 The figure is presented in the previous manuscript, in this case it diminishes the readability of the article. More details about the KEIGAFF approach as the acronym spelling may improve fruition.

The figure is deleted. The acronym is provided in the introduction (lines 63-65).

Lines 92-93 This is awkward wording.

The repetitive use of advise (advisors/advised) is replaced by other wording (lines 94-95).

Line 93 It is not clear what do you mean for evidence-based activities.

To be more consistent with the description of the intervention in the protocol paper we changed this part into ‘exchanged best-practices’ (line 96).

Lines 94-95 This statement will be more appropriate in the next section.

As suggested, we removed this part to ‘Study design’ (line 75).

Lines 100-104 These statement are inappropriate in the method section.

These statements were meant to inform the reader on the objectives of the working groups. As this may have led to misinterpretation, we removed the word ‘Ideally’ and stated that ‘The working groups were advised on implementing a comprehensive and integrated set …’ (lines 98-99).

Lines 113-120 This paragraph is not so clear and it seem s to be more pertinent with the section 2.4.

As suggested, this paragraph is removed to section 2.4 Data collection and the information is rewritten to improve clarity (lines 134-141).

Lines 123-128 To make the reading smoother this section can be combined with the previous one.

We agree that readability could be improved and we combined information and restructured information to enhance readability (lines 67-74; lines 109-129).

Lines 126-128 This is not an explanation of the setting and population.

We improved the explanation on the setting and population by using the CFIR of Damschroder et al. (2009) to explain what we mean with context and provide examples of the different type of settings (lines 115-132). More information on the study population is provided in the paragraphs on the different data collection tools. We stated that in the paragraph ‘Study setting and study population’ (line 109-129).

/ Line 129 Overall the data collection section is difficult to understand since the generic explanation of the process is not consistent with these passages that now suddenly appear.

To aid the reading on the data collection process, we added Figure 1 (line 143).

Line 130 Why are you starting from the timeline session if they are the last action you did?

As suggested, we restructured the data collection tools and present them in chronological order.

Lines 131-132 Repetitions are present.

We did not find the repetitions you are referring to, however, restructured that paragraph slightly to enhance readability (lines 174-188).

Lines 133-134 it seems a result not a method.

We agree that this seems like a result. However, we consider it best to present this information in the Methods-section and not in the Results-section, since it does not fit any of the information presented in the Results-section.

Lines 135-136 What it means?

The HP advisor also brainstormed with the other working group members on the implemented activities and perceived highlights and failures.

Line 140 This is a result.

Again, we considered this contextual information belonging to our methodology rather than a result. Results in this manuscript are fully directed towards the process of implementation and associated factors.

Line 141 Why were these interviews conducted? Who selected this purposive sample and on which criterion? How was developed the semi-structured interview guide?

The interviews were conducted to gain more insight into implementation at different levels (operational level and management level) and factors influencing implementation. (lines 191-192) This sample was chosen by the main researcher based on the diversity in implementation of the KEIGAAF approach. (lines 192-193). Relevant factors of the CFIR were chosen as guidance for the interview questions.

Line 150 This section is not clear at all. Why the minutes of working groups meeting are important? Furthermore, what are the working group meeting?

The minutes of these meetings provided in-depth information about the implementation plan from informing the principals to developing and implementing plans and the (supporting) role of the steering committee in implementation. (lines 145-152). The working group meetings are the meetings between the working group members.

Line 157 It is 2.4.4. and what do you mean for participatory observations? Line 159 The number of notes is a result and why is it relevant?

Participatory observations are observations in which the researcher is a part of practice: he/she is engaged in the intervention and while (supporting in) implementation he/she observes the process, reactions of people, physical changes, etc. Concerning the number of notes: see our previous response concerning the decision to present this in the Methods-section. The number is relevant to show the amount of data gather during intervention implementation.

Lines 163-167 This is awkward wording.

We do not fully understand the reviewer’s comment on this. What particular part of this sentence is considered as awkward wording?

Lines 168 Why the starting situation is mentioned at the end of methods section? /

As suggested, we restructured section on the data collection tools and present the tools in chronological order of use.

Lines 185 It is not clear how the codes were linked to the five domains of the CFIR.

To explain how the codes were linked, we provide examples of the inductive codes for each domain of the CFIR.

Figure 2. It is clear, only December 2016 is not readable. The problem is that the manuscript is not in line with the figure. The distinction between implementation and research is not well addressed.

Figure 2 is replaced by a more reader-friendly figure of a timeline of implementation (line 233). Besides, implementation is described more elaborate in text (lines 224-312).

Table 1. The table as currently worded is difficult to read and it does not help understanding differences between schools. You need to represent only the main different points.

Table 1 is deleted in the manuscript. Main differences are explained in text (lines 235-313).

Line 212 The remaining domain because the process domain was explained with table 1? The process domain was the last one. Be more consistent with adapted and unadapted?

To align with the order of our research questions, we first presented the implementation process before proceeding with the factors that influenced implementation. This seemed a more logical order, than starting with the factors first and ending with the implementation process.

Figure 3. It is a summary of the facilitating and hindering factors in the implementation of the KEIGAAF per domain of the CFIR. The following table is redundant with the sections developed below and did not add useful information. This scheme can be a good guide to developing the points in the manuscript but in a consistent manner (see “Wrong Interpretation of the KEIGAAF approach” in the figure and “Interpretation of the KEIGAAF approach” in the text).

Table 2 is deleted for the same reasons as Table 1: the table was too detailed and indeed redundant. We ensured that the context in Figure 3 aligned the information in text.

Line 247-266 Be more concise with the main differences between schools.

The information in this paragraph was condensed. Concerning this factor, there were no differences between schools.

Lines 297-298 You need to show only the results. You need to avoid comments in the whole results section.

We deleted or modified sentences in the Results section to avoid discussing results in this section. For example, we added ‘by the working group members’ behind ‘was considered important’ (line 348).

Lines 398-404 This section is redundant.

Since this section was considered information concerning the process it was included in the description on the implementation of KEIGAAF (the first part of the results) (lines 283-288).

Line 447 It is not clear how the reference could help in the summary of the results. /

The reference was deleted.

Box 2. HP advisors and researches are used sometimes interchangeably.

The recommendations were split into recommendations for HP advisors and recommendations for researchers (line 549).

Lines 496-497 these tools need to be clearly described in the methods section and introduced better at the beginning.

To ensure that these tools were also mentioned in the Methods section, we added ‘Initially, the working group… for this context.” (lines 175-179).

Lines 499-500 This passage is not clear in the manuscript.

We do not understand what is not clear in this passage. Could you be more specific?

Lines 501-504 It is not sufficiently clear.

The information on the flexibility in the research approach entailing that measurements were deleted and added throughout the process is elaborated some more in text (lines 559-565) and also shown in Figure 1 (line 143).

Lines 523-530 Conclusion did not give punctual answers to the main aims.

The conclusion was slightly modified to conclude the main results and provide punctual answers (lines 585-594).

References are numbered twice.

The double numbering of the references is changed. We apologize for this mistake.

Check reference 18.

The page numbers were removed and replaced by the ISBN (line 668).

Where are reported the results of the questions showed in the supplemental file 1?

The results of these questions were analysed and results are reported in text.

Supplemental file 2. The physical education duration row you do not need to repeat PE for each column.

This supplemental file is replaced by another supplemental file.

Reference:

Damschroder, L.J.; Aron, D.C.; Keith, R.E.; Kirsh, S.R.; Alexander, J.A.; Lowery, J.C. Fostering implementation of health services research findings into practice: a consolidated framework for advancing implementation science. Implement Sci 2009, 4, 50, doi:10.1186/1748-5908-4-50.

Round 2

Reviewer 1 Report

Thank you for the opportunity to review this revised manuscript. The authors have been very responsive to the prior review. The manuscript is now much improved. I have two minor comments / questions remaining:

In Figure 1, I am confused by the strikeout of the text, “team climate inventory” There were a number of barriers to implementation mentioned, but not much detail provided on “lessons learned” regarding how to overcome these barriers. I think providing some context on how to (potentially) resolve these barriers for future programs would be extremely beneficial for readers. I recommend adding this in the discussion section and potentially adding information in Box 2.

Author Response

Dear reviewer,

Thank you very much for reviewing our revised manuscript and for your compliments. We have provided our response to the two questions/comments below. The changes were highlighted in red in the manuscript. A professional editor has edited this revised version of the manuscript.

In Figure 1, I am confused by the strikeout of the text, “team climate inventory”.

We understand the confusion concerning the strikeout of the text “team climate inventory”. We decided to delete it in the figure. We wanted to show that the measurement instrument was part of the initial plan of the process evaluation and that we used the tool in the first year of implementation. However, we noticed during implementation that the tool was inappropriate for the context: the working groups were highly dynamic and follow-up of the team climate required a more or less stable team. However, since Figure 1 shows the actual measurement tools used for the process evaluation and implementation, we deleted it from the figure.

There were a number of barriers to implementation mentioned, but not much detail provided on “lessons learned” regarding how to overcome these barriers. I think providing some context on how to (potentially) resolve these barriers for future programs would be extremely beneficial for readers. I recommend adding this in the discussion section and potentially adding information in Box 2.

Thank you for the suggestion. You request some more detail on how barriers were resolved to inform intervention developers and implementers of future programs. Therefore, we added examples in the discussion regarding the role of the HP advisor in overcoming the barriers (lines 524-536). Additionally, we added more detailed recommendations concerning the supportive role of the HP advisor in the implementation of the intervention (Box 2, line 560). The changes in the box were also highlighted in red. We hope that these changes in text provide some context and will aid future programs in implementation.

Reviewer 2 Report

The manuscript has been significantly improved. The revised version improves reader-friendliness of the manuscript with a better alignment of the terminology. The actual introduction is better focused on the importance of implementing mutual adaptation interventions but the main aim actually in not clear.

The ‘Materials and Methods’ section was the one that  in the original manuscript did not enhance readability creating confusion and disorientation. The actual Materials and Methods’ section aligned the information presented in the protocol paper leading with greater clarity to reading the results.

Actually also the results section is aligned with the main research questions. The new 2 figures and the fact that the 2 tables have been deleted makes this section more usable. The information was condensed to avoid discussing results. The recommendations split into recommendations for HP advisors and

recommendations for researchers make this section more clear.

Greater consistency is needed between aims and discussion section.

Minor revisions:

Lines 19-20: repetition of the term implemented.

Lines 22-23: repetition of the term study.

In the abstract section the purpose of this article is not clearly spelled out.

Line 52: reference 17 was cited before the 16 one?

Lines 60-61: the objective of this article now is to provide an example of the implementation and evaluation of a mutual adaptation physical activity and nutrition intervention? it is completely different from the previous one. In the next sections you are talking about a process evaluation. Again, you need to be consistent.

Line 102: of 2 years.

Line 235and255and 276: add of.

Lines 278-279: repetition of activities

Line 314 : it would be better to be more concise and direct in writing the following section explanatory of figure 3

Line 403: the dot at the end of the sentence.

Author Response

Dear reviewer,

Thank you very much for reviewing our revised manuscript. We agreed with the suggestions for improvement provided for the first revision and we are happy to read that the revised manuscript is much more reader-friendly. Thank you very much for the compliments. Our response to your minor comments are provided below point-by-point.  The changes were highlighted in red in the manuscript. A professional editor has edited this revised version of the manuscript.

Greater consistency is needed between aims and discussion section.

We regret that the aims and the discussion section are not consistent with each other. To improve consistency, we have modified the sentence “we provide an example of the implementation and evaluation of a mutual adaptation physical activity and nutrition intervention that was implemented in primary schools in the Netherlands” to “we evaluated a mutual adaptation physical activity and nutrition intervention that was implemented in primary schools in the Netherlands” (lines 61-62). Additionally, we have explicitly mentioned our first and our second study aim in the discussion (line 503).

Minor revisions:

Lines 19-20: repetition of the term implemented.

We have modified the sentence slightly and removed the term ‘implementation’. (lines 20-21).

Lines 22-23: repetition of the term study.

We have changed the word ‘studied’ to ‘evaluated’. (line 21).

In the abstract section the purpose of this article is not clearly spelled out.

We have added a sentence and modified the first sentence to ensure that the purpose of the article was explained in the abstract. (lines 17-20).

Line 52: reference 17 was cited before the 16 one?

Thank you for the attentiveness. We have checked the references carefully and ensured that the references were correct. Accordingly, reference 16 and 17 were moved (lines 53, 54, 57, 139, 515, 540, 548, 551, 557).

Lines 60-61: the objective of this article now is to provide an example of the implementation and evaluation of a mutual adaptation physical activity and nutrition intervention? it is completely different from the previous one. In the next sections you are talking about a process evaluation. Again, you need to be consistent.

The objective (“we provide an example of the implementation and evaluation of a mutual adaptation physical activity and nutrition intervention that was implemented in primary schools in the Netherlands”) was not changed in this first revised version compared to the previous version. However, we do understand that this objective seems inconsistent with the research questions. Therefore, we have modified the sentence to “we evaluated a mutual adaptation physical activity and nutrition intervention that was implemented in primary schools in the Netherlands” (lines 61-62).

Line 102: of 2 years.

Again, we thank you for the attentiveness. We have added ‘of two years’. (line 103).

Line 235 and 255 and 276: add of.

As suggested, we added ‘of’ to these three subheadings. (lines 236, 256, 277).

Lines 278-279: repetition of activities

We have deleted the use of “of the activities” two times and referred to the activities by using the word ‘their’. (lines 278, 280).

Line 314 : it would be better to be more concise and direct in writing the following section explanatory of figure 3

We have deleted the first sentence of this section, because it was redundant and we have modified the second sentence slightly to make it more concise and direct. (lines 316-317).

Line 403: the dot at the end of the sentence.

We have added the forgotten dot at the end of the sentence. (line 402).